# Large Language Models Know Your Contextual Search Intent: A Prompting Framework for Conversational Search

Kelong Mao[1,2], Zhicheng Dou[1,2]*, Fengran Mo[3], Jiewen Hou[4],
Haonan Chen[1], Hongjin Qian[1]

[1]Gaoling School of Artificial Intelligence, Renmin University of China
[2]Engineering Research Center of Next-Generation Search and Recommendation, MOE
[3]Université de Montréal, Québec, Canada
[4]Institute of Computing Technology, Chinese Academy of Sciences
{mkl,dou}@ruc.edu.cn

## Abstract

Precisely understanding users' contextual search intent has been an important challenge for conversational search. As conversational search sessions are much more diverse and long-tailed, existing methods trained on limited data still show unsatisfactory effectiveness and robustness to handle real conversational search scenarios. Recently, large language models (LLMs) have demonstrated amazing capabilities for text generation and conversation understanding. In this work, we present a simple yet effective prompting framework, called *LLM4CS*, to leverage LLMs as a text-based search intent interpreter to help conversational search. Under this framework, we explore three prompting methods to generate multiple query rewrites and hypothetical responses, and propose to aggregate them into an integrated representation that can robustly represent the user's real contextual search intent. Extensive automatic evaluations and human evaluations on three widely used conversational search benchmarks, including CAsT-19, CAsT-20, and CAsT-21, demonstrate the remarkable performance of our simple LLM4CS framework compared with existing methods and even using human rewrites. Our findings provide important evidence to better understand and leverage LLMs for conversational search. The code is released at https://github.com/kyriemao/LLM4CS.

## 1 Introduction

Conversational search has been expected to be the next generation of search paradigms (Culpepper et al., 2018). It supports *search via conversation* to provide users with more accurate and intuitive search results and a much more user-friendly search experience. Unlike using traditional search engines which mainly process keyword queries, users could imagine the conversational search system as a knowledgeable human expert and directly start a

multi-turn conversation with it in natural languages to solve their questions. However, one of the main challenges for this beautiful vision is that the users' queries may contain some linguistic problems (e.g., omissions and coreference) and it becomes much harder to capture their real search intent under the multi-turn conversation context (Dalton et al., 2021; Mao et al., 2022a).

To achieve conversational search, an intuitive method known as *Conversational Query Rewriting (CQR)* involves using a rewriting model to transform the current query into a de-contextualized form. Subsequently, any ad-hoc search models can be seamlessly applied for retrieval purposes. Given that existing ad-hoc search models can be reused directly, CQR demonstrates substantial practical value for industries in quickly initializing their conversational search engines. Another type of method, *Conversational Dense Retrieval (CDR)*, tries to learn a conversational dense retriever to encode the user's real search intent and passages into latent representations and performs dense retrieval. In contrast to the two-step CQR method, where the rewriter is difficult to be directly optimized towards search (Yu et al., 2021; Mao et al., 2023a), the conversational dense retriever can naturally learn from session-passage relevance signals.

However, as conversational search sessions are much more diverse and long-tailed (Mao et al., 2022b; Dai et al., 2022; Mo et al., 2023a), existing CQR and CDR methods trained on limited data still show unsatisfactory performance, especially on more complex conversational search sessions. Many studies (Vakulenko et al., 2021b; Lin et al., 2021a; Qian and Dou, 2022; Krasakis et al., 2022) have demonstrated the performance advantages of using de-contextualized human rewrites on sessions which have complex response dependency. Also, as reported in the public TREC CAsT 2021 benchmark (Dalton et al., 2022), existing methods still suffer from significant degradation in their ef-

---
*Corresponding author.

fectiveness as conversations become longer.

Recently, large language models (LLMs) have shown amazing capabilities for text generation and conversation understanding (Brown et al., 2020; Wei et al., 2022; Thoppilan et al., 2022; Zhu et al., 2023). In the field of information retrieval (IR), LLMs have also been successfully utilized to enhance relevance modeling via various techniques such as query generation (Bonifacio et al., 2022; Dai et al., 2023), query expansion (Wang et al., 2023a), document prediction (Gao et al., 2022; Mackie et al., 2023), etc. Inspired by the strong performance of LLMs in conversation and IR, we try to investigate how LLMs can be adapted to precisely grasp users' contextual search intent for conversational search.

In this work, we present a simple yet effective prompting framework, called *LLM4CS*, to leverage LLM as a search intent interpreter to facilitate conversational search. Specifically, we first prompt LLM to generate both short query rewrites and longer hypothetical responses in multiple perspectives and then aggregate these generated contents into an integrated representation that robustly represents the user's real search intent. Under our framework, we propose three specific prompting methods and aggregation methods, and conduct extensive evaluations on three widely used conversational search benchmarks, including CAsT-19 (Dalton et al., 2020), CAsT-20 (Dalton et al., 2021), and CAsT-21 (Dalton et al., 2022)), to comprehensively investigate the effectiveness of LLMs for conversational search.

In general, our framework has two main advantages. First, by leveraging the powerful contextual understanding and generation abilities of large language models, we show that additionally generating hypothetical responses to explicitly supplement more plausible search intents underlying the short rewrite can significantly improve the search performance. Second, we show that properly aggregating multiple rewrites and hypothetical responses can effectively filter out incorrect search intents and enhance the reasonable ones, leading to better search performance and robustness.

Overall, our main contributions are:

- We propose a prompting framework and design three tailored prompting methods to leverage large language models for conversational search, which effectively circumvents the serious data scarcity problem faced by the con-

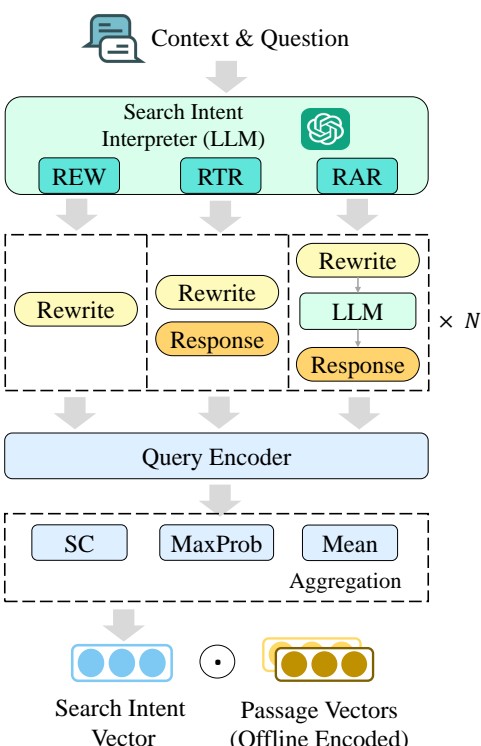

Figure 1: An overview of LLM4CS.

versational search field.

- We show that additionally generating hypothetical responses and properly aggregating multiple generated results are crucial for improving search performance.

- We demonstrate the exceptional effectiveness of LLMs for conversational search through both automatic and human evaluations, where the best method in our LLM4CS achieves remarkable improvements in search performance over state-of-the-art CQR and CDR baselines, surpassing even human rewrites.

## 2   Related Work

**Conversational Search**. Conversational search is an evolving field that involves retrieving relevant information based on multi-turn dialogues with users. To achieve conversational search, two main methods have been developed: conversational query rewriting and conversational dense retrieval. Conversational query rewriting converts the conversational search problem into an ad-hoc search problem by reformulating the search session into a standalone query rewrite. Existing methods try to select useful tokens from the conversation context (Voskarides et al., 2020; Lin et al., 2021b)

or train a generative rewriter based on the pairs of sessions and rewrites (Lin et al., 2020; Yu et al., 2020; Vakulenko et al., 2021a). To make the rewriting process aware of the downstream retrieval process, some studies propose to adopt reinforcement learning (Wu et al., 2022; Chen et al., 2022) or enhance the learning of rewriter with ranking signals (Mao et al., 2023a; Mo et al., 2023a). On the other hand, conversational dense retrieval (Yu et al., 2021) directly encodes the whole conversational search session to perform end-to-end dense retrieval. Existing methods mainly try to improve the session representation through context denoising (Mao et al., 2022a; Krasakis et al., 2022; Mo et al., 2023b; Mao et al., 2023b), data augmentation (Lin et al., 2021a; Mao et al., 2022b; Dai et al., 2022), and hard negative mining (Kim and Kim, 2022).

**IR with LLMs**. Due to the revolutionary natural language understanding and generation abilities, LLMs are attracting more and more attention from the IR community. LLMs have been leveraged to enhance the relevance modeling of retrieval through query generation (Bonifacio et al., 2022; Jeronymo et al., 2023; Dai et al., 2023), query expansion (Wang et al., 2023a), document prediction (Gao et al., 2022; Mackie et al., 2023), etc. Besides, Shen et al. (2023) proposed to first use the retriever to enhance the generation of LLM and then use the generated content to augment the original search query for better retrieval. Ziems et al. (2023) treated LLM as a built-in search engine to retrieve documents based on the generated URL. There are also some works leveraging LLM to perform re-ranking (Sun et al., 2023; Jiang et al., 2023). Different from previous studies, in this paper, we propose the LLM4CS framework that focuses on studying how LLM can be well utilized to capture the user's contextual search intent to facilitate conversational search.

# 3 LLM4CS: Prompting Large Language Models for Conversational Search

In this section, we introduce our LLM4CS framework, which leverages LLM as a text-based search intent interpreter to facilitate conversational search. Figure 1 shows an overview of LLM4CS. In the following, we first describe our task formulation of conversational search, and then we elaborate on the specific prompting methods and aggregation methods integrated into the framework. Finally, we introduce the retrieval process.

## 3.1 Task Formulation

We focus on the task of conversational passage retrieval, which is the crucial first step of conversational search that helps the model access the right evidence knowledge. Given the user query $q^t$ and the conversation context $C^t = (q^1, r^1, ..., q^{t-1}, r^{t-1})$ of the current turn $t$, where $q^i$ and $r^i$ denote the user query and the system response of the historical $i$-th turn, our goal is to retrieve passages that are relevant to satisfy the user's real search intent of the current turn.

## 3.2 Prompting Methods

The prompt follows the formulation of *[Instruction, Demonstrations, Input]*, where *Input* is composed of the query $q^t$ and the conversation context $C^t$ of the current turn $t$. Figure 4 shows a general illustration of the prompt construction.[1] Specifically, we design and explore three prompting methods, including *Rewriting (REW)*, *Rewriting-Then-Response (RTR)*, and *Rewriting-And-Response (RAR)*, in our LLM4CS framework.

### 3.2.1 Rewriting Prompt (REW)

In this prompting method, we directly treat LLM as a well-trained conversational query rewriter and prompt it to generate rewrites. Only the red part of Figure 4 is enabled. Although straightforward, we show in Section 4.5 that this simple prompting method has been able to achieve quite a strong search performance compared to existing baselines.

### 3.2.2 Rewriting-Then-Response (RTR)

Recently, a few studies (Mao et al., 2021; Gao et al., 2022; Yu et al., 2023; Mackie et al., 2023) have shown that generating hypothetical responses for search queries can often bring positive improvements in retrieval performance. Inspired by them, in addition to prompting LLM to generate rewrites, we continue to utilize the generated rewrites to further prompt LLM to generate hypothetical responses that may contain relevant information to answer the current question. The orange part and the blue part of Figure 4 are enabled. Specifically, we incorporate the pre-generated rewrite (i.e., the orange part) into the *Input* field of the prompt and

---

[1]We put this figure in Appendix A due to the space limitation. See our open-sourced code for the full prompt of each prompting method.

then prompt LLM to generate informative hypothetical responses by referring to the rewrite.

### 3.2.3 Rewriting-And-Response (RAR)

Instead of generating rewrites and hypothetical responses in a two-stage manner, we can also generate them all at once with the red part and the blue part of Figure 4 being enabled. We try to explore whether this one-stage generation could lead to better consistency and accuracy between the generated rewrites and responses, compared with the two-step RTR method.

### 3.2.4 Incorporating Chain-of-Thought

Chain-of-thought (CoT) (Wei et al., 2020) induces the large language models to decompose a reasoning task into multiple intermediate steps which can unlock their stronger reasoning abilities. In this work, we also investigate whether incorporating the chain-of-thought of reasoning the user's real search intent could improve the quality of rewrite and response generation.

Specifically, as shown in the green part of Figure 4, we manually write the chain-of-thought for each turn of the demonstration, which reflects how humans infer the user's real search intent of the current turn based on the historical conversation context. When generating, we instruct LLM to first generate the chain-of-thought before generating rewrites (and responses). We investigate the effects of our proposed CoT tailored to the reasoning of contextual search intent in Section 4.6.

### 3.3 Content Aggregation

After prompting LLM multiple times to generate multiple rewrites and hypothetical responses, we then aggregate these generated contents into an integrated representation to represent the user's complete search intent for search. Let us consider that we have generated $N$ query rewrites $\mathcal{Q} = (\hat{q}_1, ..., \hat{q}_N)$ and $M$ hypothetical responses $\mathcal{R} = (\hat{r}_{i1}, ..., \hat{r}_{iM})$ for each rewrite $\hat{q}_i$, sorted by their generation probabilities from high to low[2]. Note that in RAR prompting, the rewrites and the hypothetical responses are always generated in pairs (i.e., $M = 1$). While in RTR prompting, one rewrite can have $M$ hypothetical responses since they are generated in a two-stage manner. Next, we utilize a dual well-trained ad-hoc retriever[3] $f$

---

[2]That is, the generation probability orders are: $P(\hat{q}_1) \geq ... \geq P(\hat{q}_N)$ and $P(\hat{r}_{i1}) \geq ... \geq P(\hat{r}_{iM})$.

[3]The parameters of the query encoder and the passage encoder are shared.

(e.g, ANCE (Xiong et al., 2021)) to encode each of them into a high-dimensional intent vector and aggregate these intent vectors into one final search intent vector $\mathbf{s}$. Specifically, we design and explore the following three aggregation methods, including *MaxProb*, *Self-Consistency (SC)*, and *Mean*, in our LLM4CS framework.

### 3.3.1 MaxProb

We directly use the rewrite and the hypothetical response that have the highest generation probabilities. Therefore, compared with the other two aggregation methods that will be introduced later, MaxProb is highly efficient since it actually does not require multiple generations.

Formally, for REW prompting:

$$\mathbf{s} = f(\hat{q}_1). \tag{1}$$

For the RTR and RAR prompting methods, we mix the rewrite and hypothetical response vectors:

$$\mathbf{s} = \frac{f(\hat{q}_1) + f(\hat{r}_{11})}{2}. \tag{2}$$

### 3.3.2 Self-Consistency (SC)

The multiple generated rewrites and hypothetical responses may express different search intents but only some of them are correct. To obtain a more reasonable and consistent search intent representation, we extend the self-consistency prompting method (Wang et al., 2023b), which was initially designed for reasoning tasks with predetermined answer sets, to our contextual search intent understanding task, which lacks a fixed standard answer. To be specific, we select the intent vector that is the most similar to the cluster center of all intent vectors as the final search intent vector, since it represents the most popular search intent overall.

Formally, for REW prompting:

$$\hat{\mathbf{q}}^* = \frac{1}{N} \sum_{i=1}^{N} f(\hat{q}_i), \tag{3}$$

$$\mathbf{s} = \arg\max_{f(\hat{q}_i)} f(\hat{q}_i)^\top \cdot \hat{\mathbf{q}}^*, \tag{4}$$

where $\hat{\mathbf{q}}^*$ is the cluster center vector and $\cdot$ denotes the dot product that measures the similarity.

For RTR prompting, we first select the intent vector $f(\hat{q}_k)$ and then select the intent vector $f(\hat{r}_{kz})$ from all hypothetical responses generated based on

the selected rewrite $\hat{q}_k$:

$$k \;=\; \arg\max_i f(\hat{q}_i)^\top \cdot \hat{\mathbf{q}}^*, \tag{5}$$

$$\hat{\mathbf{r}}_k^* \;=\; \frac{1}{M}\sum_{j=1}^{M} f(\hat{r}_{kj}), \tag{6}$$

$$z \;=\; \arg\max_j f(\hat{r}_{kj})^\top \cdot \hat{\mathbf{r}}_k^*, \tag{7}$$

$$\mathbf{s} \;=\; \frac{f(\hat{q}_k) + f(\hat{r}_{kz})}{2}, \tag{8}$$

where $k$ and $z$ are the finally selected indexes of the rewrite and the response, respectively.

The aggregation for RAR prompting is similar to RTR prompting, but it does not need response selection since there is only one hypothetical response for each rewrite:

$$\mathbf{s} \;=\; \frac{f(\hat{q}_k) + f(\hat{r}_{k1})}{2}. \tag{9}$$

### 3.3.3 Mean

We average all the rewrite vectors and the corresponding hypothetical response vectors.

For REW prompting:

$$\mathbf{s} = \frac{1}{N}\sum_{i=1}^{N} f(\hat{q}_i). \tag{10}$$

For the RTR and RAR prompting methods:

$$\mathbf{s} = \frac{\sum_{i=1}^{N}[f(\hat{q}_i) + \sum_{j=1}^{M} f(\hat{r}_{ij})]}{N * (1 + M)}. \tag{11}$$

Compared with MaxProb and Self-Consistency, the Mean aggregation comprehensively considers more diverse search intent information from all sources. It leverages the collaborative power to enhance the popular intents, but also supplements plausible intents that are missing in a single rewrite or a hypothetical response.

### 3.4 Retrieval

All candidate passages are encoded into passage vectors using the same retriever $f$. At search time, we return the passages that are most similar to the final search intent vector $\mathbf{s}$ as the retrieval results.

## 4 Experiments

### 4.1 Datasets and Metrics

We carry out extensive experiments on three widely used conversational search datasets: CAsT-19 (Dalton et al., 2020), CAsT-20 (Dalton et al., 2021), and

| Dataset | CAsT-19 | CAsT-20 | CAsT-21 |
|---|---|---|---|
| # Conversations | 20 | 25 | 18 |
| # Turns (Sessions) | 173 | 208 | 157 |
| # Passages/Docs | | 38M | 40M |

Table 1: Statistics of the three CAsT datasets.

CAsT-21 (Dalton et al., 2022), which are curated by the human experts of TREC Conversational Assistance Track (CAsT). Each CAsT dataset has dozens of information-seeking conversations comprising hundreds of turns. CAsT-19 and CAsT-20 share the same retrieval corpora while CAsT-21 has a different one. In contrast, CAsT-20 and CAsT-21 have a more complex session structure than CAsT-19 as their questions may refer to previous responses. All three datasets provide human rewrites and passage-level (or document-level) relevance judgments labeled by TREC experts. Table 1 summarizes the basic dataset statistics.[4]

Following previous work (Dalton et al., 2020, 2021; Yu et al., 2021; Mao et al., 2022a), we adopt Mean Reciprocal Rank (MRR), NDCG@3, and Recall@100 as our evaluation metrics and calculate them using `pytrec_eval` tool (Van Gysel and de Rijke, 2018). We deem relevance scale $\geq 2$ as positive for MRR on CAsT-20 and CAsT-21. For CAsT-21, we split the documents into passages and score each document based on its highest-scored passage (i.e., *MaxP* (Dai and Callan, 2019)). We conduct the statistical significance tests using paired t-tests at $p < 0.05$ level.

### 4.2 Implementation details

We use the OpenAI *gpt3.5-turbo-16k* as our LLM. The decoding temperature is set to 0.7. We randomly select three conversations from the CAsT-22[5] dataset for demonstration. CAsT-22 is a new conversational search dataset also proposed by TREC CAsT, but only its conversations are released[6] and the relevance judgments have not been made public. Therefore, it cannot be used for evaluation and we just use it for demonstration. For REW prompting, we set $N = 5$. For RTR prompting, we set $N = 1$ and $M = 5$. For RAR prompting, we set $N = 5$, and $M$ is naturally set to 1. Following previous studies (Yu et al., 2021; Mao

---

[4]Only the turns that have relevance labels are counted.
[5]https://github.com/daltonj/treccastweb/tree/master/2022
[6]Until the submission deadline of EMNLP 2023.

et al., 2022a,b; Mo et al., 2023a), we adopt the ANCE (Xiong et al., 2021) checkpoint pre-trained on the MSMARCO dataset as our ad-hoc retriever $f$. We uniformly truncate the lengths of queries (or rewrites), passages, and hypothetical responses into 64, 256, and 256.

## 4.3 Baselines

We compare our few-shot LLM4CS against the following six conversational search systems:

(1) **T5QR** (Lin et al., 2020): A T5 (Raffel et al., 2020)-based conversational query rewriter trained with the human rewrites.

(2) **ConvDR** (Yu et al., 2021): A conversational dense retriever fine-tuned from an ad-hoc retriever by mimicking the representations of human rewrites.

(3) **COTED** (Mao et al., 2022a): An improved version of ConvDR (Yu et al., 2021) which incorporates a curriculum learning-based context denoising objective.

(4) **ZeCo** (Krasakis et al., 2022): A variant of ColBERT (Khattab and Zaharia, 2020) that matches only the contextualized terms of the current query with passages to perform zero-shot conversational search.

(5) **CRDR** (Qian and Dou, 2022): A conversational dense retrieval method where the dense retrieval part is enhanced by the distant supervision from query rewriting in a unified framework.

(6) **ConvGQR** (Mo et al., 2023a): A query reformulation framework that combines query rewriting with generative query expansion.

T5QR, CRDR, and ConvGQR are trained on the training sessions of QReCC (Anantha et al., 2021), which is a large-scale conversational question answering dataset. The performances of ConvDR and COTED are reported in the few-shot setting using 5-fold cross-validation according to their original papers. We also present the performance of using human rewrites for reference. Note that the same ANCE checkpoint is used to perform dense retrieval for all baselines except ZeCo to ensure fair comparisons.

## 4.4 Main Results

The overall performance comparisons are presented in Table 2. The reported performance of LLM4CS results from the combination of the RAR prompting method, the Mean aggregation method, and our tailored CoT, which shows to be the most effective combination. We thoroughly investigate the effects

of using different prompting and aggregation methods in Section 4.5 and investigate the effects of the incorporation of CoT in Section 4.6.

From Table 2, we observe that LLM4CS outperforms all the compared baselines in terms of search performance. Specifically, LLM4CS exhibits a relative improvement of over 18% compared to the second-best results on the more challenging CAsT-20 and CAsT-21 datasets across all metrics. In particular, even compared to using human rewrites, our LLM4CS can still achieve better results on most metrics, except for the Recall@100 of CAsT-19 and NDCG@3 of CAsT-21. These significant improvements, which are unprecedented in prior research, demonstrate the strong superiority of our LLM4CS over existing methods and underscore the vast potential of using large language models for conversational search.

## 4.5 Effects of Different Prompting Methods and Aggregation Methods

We present a comparison of NDCG@3 performance across various prompting and aggregation methods (excluding the incorporation of CoT) in Table 3. Our findings are as follows:

First, the RAR and RTR prompting methods clearly outperform the REW prompting, demonstrating that the generated hypothetical responses can effectively supplement the short query rewrite to improve retrieval performance. However, even the simple REW prompting can also achieve quite competitive performance compared to existing baselines, particularly on the more challenging CAsT-20 and CAsT-21 datasets, where it shows significant superiority (e.g., 0.380 vs. 0.350 on CAsT-20 and 0.465 vs. 0.385 on CAsT-21). These positive results further highlight the significant advantages of utilizing LLM for conversational search.

Second, in terms of aggregation methods, both Mean and SC consistently outperform MaxProb. These results indicate that depending solely on the top prediction of the language model may not provide sufficient reliability. Instead, utilizing the collective strength of multiple results proves to be a better choice. Additionally, we observe that the Mean aggregation method, which fuses all generated contents into the final search intent vector (Equation 11), does not consistently outperform SC (e.g., on CAsT-20), which actually only fuses one rewrite and one response (Equation 8). This suggests that taking into account more generations

| System | CAsT-19 | | | CAsT-20 | | | CAsT-21 | | |
|---|---|---|---|---|---|---|---|---|---|
| | MRR | NDCG@3 | R@100 | MRR | NDCG@3 | R@100 | MRR | NDCG@3 | R@100 |
| Conversational Dense Retrieval | | | | | | | | | |
| ConvDR | 0.740 | 0.466 | 0.362 | 0.510 | 0.340 | 0.345 | 0.573 | 0.385 | 0.483 |
| COTED | 0.769 | 0.478 | 0.367 | 0.491 | 0.342 | 0.340 | 0.565 | 0.371 | 0.485 |
| ZeCo | - | 0.238‡ | 0.216‡ | - | 0.176‡ | 0.200‡ | - | 0.234‡ | 0.267‡ |
| CRDR | 0.765 | 0.472 | 0.357 | 0.501 | 0.350 | 0.313 | 0.474 | 0.342 | 0.380 |
| Conversational Query Rewriting | | | | | | | | | |
| T5QR | 0.701 | 0.417 | 0.332 | 0.423 | 0.299 | 0.353 | 0.469 | 0.330 | 0.408 |
| ConvGQR | 0.708 | 0.434 | 0.336 | 0.465 | 0.331 | 0.368 | 0.433 | 0.273 | 0.330 |
| LLM4CS | **0.776**† | **0.515**† | **0.372**† | **0.615**† | **0.455**† | **0.489**† | **0.681**† | **0.492**† | **0.614**† |
| Human | 0.740 | 0.461 | 0.381 | 0.591 | 0.422 | 0.465 | 0.680 | 0.502 | 0.590 |
| RI-H | +4.9% | +11.7% | -2.4% | +4.1% | +7.8% | +5.2% | +0.1% | -2.0% | +4.1% |
| RI-2nd-Best | +0.9% | +7.7% | +1.4% | +20.6% | +30.0% | +32.9% | +18.8% | +27.8% | +26.6% |

Table 2: Overall performance comparisons. ‡ denotes the results are replicated from their original paper. † denotes LLM4CS (RAR + Mean + CoT) significantly outperforms all the compared baselines (except ZeCo) in the $p < 0.05$ level. The best results are bold and the second-best results are underlined. *Human* denotes using human rewrites. *RI-H* and *RI-2nd-Best* are the relative improvements over Human and the second-best results, respectively.

| Aggregation | CAsT-19 | | | CAsT-20 | | | CAsT-21 | | |
|---|---|---|---|---|---|---|---|---|---|
| | REW | RTR | RAR | REW | RTR | RAR | REW | RTR | RAR |
| MaxProb | 0.441 | 0.459 | 0.464 | 0.356 | 0.415 | 0.430 | 0.407 | 0.469 | 0.462 |
| SC | 0.449 | 0.466 | 0.476 | 0.362 | 0.432 | **0.444** | 0.445 | 0.473 | 0.469 |
| Mean | 0.447 | 0.464 | **0.488** | 0.380 | 0.425 | 0.442 | 0.465 | **0.481** | 0.478 |
| Previous SOTA | | 0.478 | | | 0.350 | | | 0.385 | |
| Human | | 0.461 | | | 0.422 | | | 0.502 | |

Table 3: Performance comparisons with respect to NDCG@3 using different prompting and aggregation methods. The best combination on each dataset is bold.

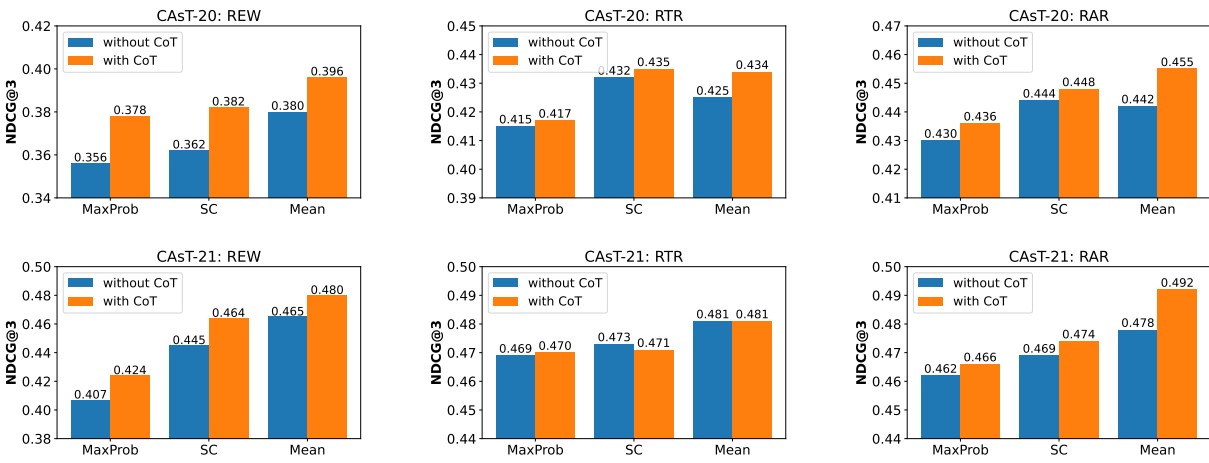

Figure 2: NDCG@3 comparisons between incorporating our tailored CoT or not across different prompting and aggregation methods on CAsT-20 and CAsT-21 datasets.

may not always be beneficial, and a careful selection among them could be helpful to achieve improved results.

### 4.6 Effects of Chain-of-Thought

We show the ablation results of our tailored chain-of-thought in Figure 2. We also provide a real example to show how our CoT takes effect in Appendix B.1. From the results, we observe that:

Incorporating our chain-of-thought into all prompting and aggregation methods generally improves search performance. This demonstrates the efficacy of our chain-of-thought in guiding the large language model towards a correct understanding of

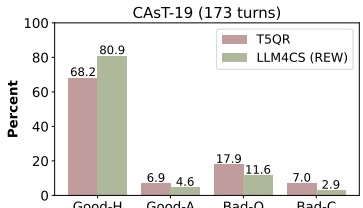 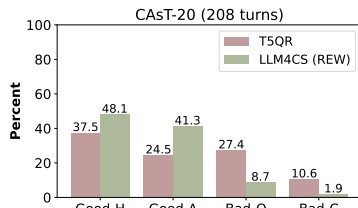 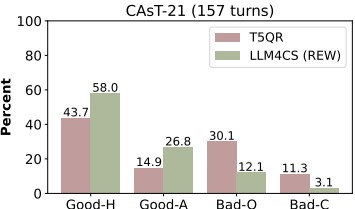

Figure 3: Human evaluation results for LLM4CS (REW + MaxProb) and T5QR on the three CAsT datasets.

the user's contextual search intent.

In contrast, the improvements are particularly notable for the REW prompting method compared to the RTR and RAR prompting methods. It appears that the introduction of multiple hypothetical responses diminishes the impact of the chain-of-thought. This could be attributed to the fact that including multiple hypothetical responses significantly boosts the quality and robustness of the final search intent vector, thereby reducing the prominence of the chain-of-thought in enhancing search performance.

## 5 Human Evaluation

The retrieval performance is influenced by the ad-hoc retriever used, which implies that automatic search evaluation metrics may not fully reflect the model's capability to understand contextual search intent. Sometimes, two different rewrites can yield significantly different retrieval scores, even though they both accurately represent the user's real search intent. To better investigate the contextual search intent understanding ability of LLM, we perform a fine-grained human evaluation on the rewrites generated by our LLM4CS (REW + MaxProb).

Specifically, we manually compare each model's rewrite with the corresponding human rewrite and label it with one of the following four categories: (1) *Good-H*: The model's rewrite is nearly the same as the human rewrite. (2) *Good-A*: The expression of the model's rewrite is different from the human rewrite but it also successfully conveys the user's real search intent. (3) *Bad-C*: the rewrite has coreference errors. (4) *Bad-O*: the rewrite omits important contextual information or has other types of errors. Furthermore, we apply the same principle to label the rewrites of T5QR for comparison purposes. A few examples of such categorization are presented in Appendix B.2.

The results of the human evaluation are shown in Figure 3, where we observe that:

(1) From a human perspective, 85.5%, 89.4%, and 84.8% of the rewrites of LLM4CS successfully convey the user's real search intent for CAsT-19, CAsT-20, and CAsT-21, respectively. In contrast, the corresponding percentages for T5QR are merely 75.1%, 62.0%, and 58.6%. Such a high rewriting accuracy of LLM4CS further demonstrates the strong ability of LLM for contextual search intent understanding.

(2) In the case of CAsT-20 and CAsT-21, a significantly higher percentage of rewrites are labeled as *Good-A*, in contrast to CAsT-19, where the majority of good rewrites closely resemble the human rewrites. This can be attributed to the higher complexity of the session structure and questions in CAsT-20 and CAsT-21 compared to CAsT-19, which allows for greater freedom in expressing the same search intent.

(3) The rewrites generated by LLM4CS exhibit coreference errors in less than 3% of the cases, whereas T5QR's rewrites contain coreference errors in approximately 10% of the cases. This observation highlights the exceptional capability of LLM in addressing coreference issues.

## 6 Conclusion

In this paper, we present a simple yet effective prompting framework (i.e., LLM4CS) that leverages LLMs for conversational search. Our framework generates multiple query rewrites and hypothetical responses using tailored prompting methods and aggregates them to robustly represent the user's contextual search intent. Through extensive automatic and human evaluations on three CAsT datasets, we demonstrate its remarkable performance for conversational search. Our study highlights the vast potential of LLMs in conversational search and takes an important initial step in advancing this promising direction. Future research will focus on refining and extending the LLM4CS framework to explore better ways of generation to facilitate search, improving aggregation techniques, optimizing the LLM-retriever interaction, and incorporating reranking strategies.

## Limitations

Our work shows that generating multiple rewrites and hypothetical responses and properly aggregating them can effectively improve search performance. However, this requires invoking LLM multiple times, resulting in a higher time cost for retrieval. Due to the relatively high generation latency of LLM, the resulting query latency would be intolerable for users when compared to conventional search engines. A promising approach is to design better prompts capable of obtaining all informative content in one generation, thereby significantly improving query latency. Another limitation is that, similar to the typical disadvantages of CQR methods, the generation process of LLM lacks awareness of the downstream retrieval process. Exploring the utilization of ranking signals to enhance LLM generation would be a compelling direction for future research of conversational search.

## Acknowledgement

This work was supported by the National Natural Science Foundation of China No. 62272467, Public Computing Cloud, Renmin University of China, and Intelligent Social Governance Platform, Major Innovation & Planning Interdisciplinary Platform for the "Double-First Class" Initiative, Renmin University of China, and the Outstanding Innovative Talents Cultivation Funded Programs 2024 of Renmin University of China. The work was partially done at Beijing Key Laboratory of Big Data Management and Analysis Methods.

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

## Appendix

## A  Prompt of LLM4CS

Figure 4 shows a general illustration of the prompt of LLM4CS. The prompt consist of three parts, which are *Instruction*, *Demonstration*, and *Input*. The red part is for REW prompting, the blue part is for the RTR and RAR promptings, and the orange part is for RTR prompting. The green part is for our designed chain-of-thought.

## B  Case Study

### B.1  Examples of Chain-of-Thought

The example in Table 4 shows how our CoT takes effect. The ***CoT*** and ***Our Rewrite*** fields are generated by LLM4CS (REW + CoT). We can find that the generated CoT effectively illustrates the rationale behind the rewriting process. Please refer to our anonymously open-sourced repository for more examples.

### B.2  Examples of Human Evaluation

An example for each category is shown in Table 5. Please refer to our anonymously open-sourced repository for more examples.

| | |
|---|---|
| Instruction | For an information-seeking dialog, please help reformulate the question into rewrite that can fully express the user's information needs without the need of context, but also generate an informative response to answer the question. |
| Demonstration | I will give you several example multi-turn dialogs, where each turn contains a question, a response, and a rewrite. The rewrite part begins with a sentence explaining the reason for the rewrite.

**Example #1:**
*Question*: What should I consider when buying a phone?
*Rewrite*: This is the first turn. So the question should be rewritten as: What should I consider when buying a phone?
*Response*: The design of the phone and the overall …

*Question*: Cool. Which one would you recommend?
*Rewrite*: Based on Turn 1, you are inquiring about what should be considered when buying a phone. So the question should be rewritten as: Cool. Which smartphone would you recommend for me?
*Response*: Just because a phone has everything…

…

**Example #2:**
… |
| Input | Your Task (only questions and responses are given):

**Context**:

*Question*: What was the basis of the Watergate scandal?
*Response*: ...

*Question*: …
*Response*: …

…

**Current Question**: So what happened to Nixon?
**Rewrite**: So what happened to Nixon after the events of the Watergate scandal?

(Now, you should give me the rewrite and an informative response of the **Current Question** based on the **Context**. The output format should always be: Rewrite: $Reason. So the question should be rewritten as: $Rewrite\nResponse: $Response.  Go ahead!) |
| Model Output | *Rewrite*: Based on all previous turns, Nixon was badly involved in the Watergate scandal. So the question should be rewritten as: So what happened to Nixon after the events of the Watergate scandal?
*Response*: With the mounting evidence and loss… |

Figure 4: A general illustration of the prompt of LLM4CS. The prompt consist of three parts, i.e., *Instruction*, *Demonstration*, and *Input*. The red part is for REW prompting, the blue part is for the RTR and RAR promptings, and the orange part is for RTR prompting. The green part is for our designed chain-of-thought.

| CAsT-20 (Conversation #84) |
|---|

**Turn 1**

*Question*: How does seed investment work for startups?
*Response*: Seed money, sometimes known as seed funding or seed capital, is a form of securities offering in which an investor invests capital in exchange for an equity stake in ...
*Human Rewrite*: How does seed investment work for startups?
*CoT*: This is the first turn.
*Our Rewrite*: How does seed investment work for startups?

**Turn 2**

*Question*: Tell me more about angel rounds.
*Response*: Tell Me More was a National Public Radio interview show hosted by journalist Michel Martin ... offering sample segments, and soliciting user feedback. It began broadcasting on April 30, 2007.
*Human Rewrite*: Tell me more about angel investment rounds.
*CoT*: Based on the current question and the previous response, we are comparing angel rounds with crowdfunding.
*Our Rewrite*: Tell me more about angel rounds in seed investment for startups.

**Turn 3**

*Question*: How does it compare with crowdfunding?
*Response*: For crowdfunding of equity stock purchases, there is some research in social psychology that indicates that ... Managing communications with a large number of possibly disappointed investors and supporters can be a substantial, and potentially diverting, task.
*Human Rewrite*: How does an angel investment round compare with crowdfunding?
*CoT*: Based on the current question and the previous response, we are comparing angel rounds with crowdfunding.
*Our Rewrite*: How does angel funding compare to crowdfunding for startups?

**Turn 4**

*Question*: How much is a typical A series round?
*Response*: OpenGov has raised $47 million from venture capital and angel investors. A $3 million Series A round in 2012 included venture funds 8VC, Founder Collective, Valiant Capital, and a number of high-profile ...
*Human Rewrite*: How much is a typical A series investment round?
*CoT*: Based on the previous turns, we are talking about different types of funding for startups such as seed investment, angel rounds, and crowdfunding.
*Our Rewrite*: How much funding is typically raised in an A series round for startups?

**Turn 5**

*Question*: What are the risks of investing?
*Response*: Angel investors invest in early stage or start-up companies in exchange for an equity ownership interest. Angel ... Here is what angels particularly care about: The quality, passion, commitment, and integrity of the founders.
*Human Rewrite*: What are the risks of investing in startups?
*CoT*: Based on the previous turns, we are discussing different aspects of startup investments.
*Our Rewrite*: What are the risks associated with investing in startups?

**Turn 6**

*Question*: What are the different types of instruments used by VCs?
*Response*: ...
*Human Rewrite*: What are the different types of instruments used by VCs?
*CoT*: The user is asking about the different types of instruments used by venture capitalists.
*Our Rewrite*: What are the various types of instruments employed by venture capitalists (VCs)?

Table 4: An example to show how our CoT takes effect. The *CoT* and *Our Rewrite* fields are generated by LLM4CS (REW + CoT).

| Good-H |
|---|
| *Question*: How about replacing it instead? |
| *Human Rewrite*: How much does it cost to replace a garage door opener? |
| *Our Rewrite*: What is the cost of replacing a garage door opener? |
| **Good-A** |
| *Question*: What's important for me to know about their safety? |
| *Human Rewrite*: What's important for me to know about the safety of smart garage door openers? |
| *Our Rewrite*: What are the safety features that I should consider when choosing a garage door opener? |
| **Bad-O** |
| *Question*: Are there any famous foods? |
| *Human Rewrite*: Are there any famous foods in Washington D.C.? |
| *Our Rewrite*: Are there any famous foods? |
| **Bad-C** |
| *Question*: What is its main economic activity? |
| *Human Rewrite*: What is the main economic activity of Salt Lake City? |
| *Our Rewrite*: What is the main economic activity in Utah? |

Table 5: Examples of the four categories in human evaluation.