# OpenReview forum: "Large Language Models Know Your Contextual Search Intent: A Prompting Framework for Conversational Search"
_EMNLP/2023/Conference — EMNLP 2023 Findings_

### Official Review · Reviewer_ARYj · 2023-08-01

**Soundness:** 4

**Excitement:**

4: Strong: This paper deepens the understanding of some phenomenon or lowers the barriers to an existing research direction.

**Missing References:**

1. Missing MSMARCO  citation @article{Campos2016MSMA,
  title={MS MARCO: A Human Generated MAchine Reading COmprehension Dataset},
  author={Daniel Fernando Campos and Tri Nguyen and Mir Rosenberg and Xia Song and Jianfeng Gao and Saurabh Tiwary and Rangan Majumder and Li Deng and Bhaskar Mitra},
  journal={ArXiv},
  year={2016},
  volume={abs/1611.09268},
  url={https://api.semanticscholar.org/CorpusID:1289517}
}
2.

**Paper Topic And Main Contributions:**

The authors introduce LLM4CS and conversation query rewriting and response aggregation framework, which improves the performance of conversational information retrieval by a wide margin.
The authors perform an extensive and systematic evaluation on three widely used conversational search benchmarks, CAsT-19, CAsT-20, and CAsT-21, demonstrating how their methodology consistently outperforms previous SOTA by a wide margin,

The method is simple yet effective and has an LLM that generates Rewriting (REW), Rewriting-Then-Response (RTR), and Rewriting And-Response (RAR) which when paired with an ANCE-based retriever, leads to impressive improvements in retrieval accuracy.

**Questions For The Authors:**

1. The anonymized github link is not functional please fix.

**Reasons To Accept:**

1. LLM4CS works incredibly well. It outperforms all prior CQR and CDR methods on a wide variety of benchmarks and by a significant margin.
2. LLM4CS has three methods REW RTR RAR, all of which work better than prior SOTA even before they are aggregated.

**Reasons To Reject:**

1. There is no discussion about the impact of using LLM ( gpt3.5-turbo-16k) for all the LLM4CS prompts before retrieval. While the performance improvements are significant, it is difficult to understand the tradeoffs when it is unclear how slow the LLM4CS system is nor how well it works with a nonclosed source model. Ideally, I would like to see both information on the efficiency of LM4CS relative to other methods and some validation on how model size or scale impacts the performance of LLM4CS
2. There is no discussion on how the LLM4CS method works with any other forms of retrieval, such as traditional keyword retrieval (e.g. BM25)

**Reproducibility:**

4: Could mostly reproduce the results, but there may be some variation because of sample variance or minor variations in their interpretation of the protocol or method.

**Reviewer Confidence:**

4: Quite sure. I tried to check the important points carefully. It's unlikely, though conceivable, that I missed something that should affect my ratings.

---

### Official Review · Reviewer_Un6E · 2023-08-02

**Soundness:** 3

**Excitement:**

2: Mediocre: This paper makes marginal contributions (vs non-contemporaneous work), so I would rather not see it in the conference.

**Paper Topic And Main Contributions:**

This paper aims to enhance the performance of conversational search by employing a Large Language Model (LLM) to provide additional information in the form of rewritten queries and responses.

The main contributions of this paper lie in the domain of NLP engineering experiments. The proposed framework, LLM4CS, demonstrates significant improvements in search performance compared to baseline methods. Notably, LLM4CS outperforms baselines on the CAsT-20 and CAsT-21 datasets. However, it is important to acknowledge that the generalizability and robustness of the framework are somewhat limited due to the use of only one LLM in the experiments and the relatively small dataset size.

Overall, while the contributions are valuable, they may be more suitable for a shorter paper rather than a regular long paper.

**Questions For The Authors:**

1. Why are the results for CRDR and ConvGQR on the CAsT-21 dataset not shown in Table 2?
2. Could you explain why LLM4CS achieves a larger lift on the CAsT-20 dataset compared to the CAsT-19 dataset?
3. How does the framework proposed in this paper differ from other frameworks that utilize prompts to generate and encode content using LLMs for various tasks?

**Reasons To Accept:**

1. This paper presents a straightforward yet effective prompting framework, LLM4CS, which leverages a Large Language Model for conversational search. The method combines Conversational Query Rewriting (CQR) and Conversational Dense Retrieval (CDR) techniques, rewriting queries to obtain additional information and then encoding it into dense vectors.
2. LLM4CS demonstrates remarkable performance on three CAsT datasets, particularly excelling on the CAsT-20 and CAsT-21 datasets.
3. The paper conducts extensive experiments to analyze the impact of different prompting methods and content aggregation techniques.

**Reasons To Reject:**

1. The novelty of the proposed framework is somewhat limited. Utilizing pre-trained LLMs to provide additional information may intuitively yield useful results, but the paper lacks novel methods or unexpected findings. It is not entirely clear how the content generated by LLMs leads to performance improvement.
2. The experiments conducted in the paper may not sufficiently verify the generalizability of the framework. Only one LLM, gpt-3.5-turbo, was utilized. One concern is whether the LLM used (gpt-3.5-turbo) might have seen the dataset used for the experiment, potentially leading to data leakage and affecting the generalizability of the results. It is recommended to conduct experiments with at least two different open-source LLMs with varying parameter scales. Moreover, there should be an analysis of the impact of different text encoders.
3. The proposed framework seems to be a combination of "CQR" and "CDR," i.e. rewriting and then encoding dense vectors. The experiments center around 3 "CQR" methods and 3 "CDR" methods. However, there is no result of the baseline method that combines CQR and CDR. As far as I know, there is a hybrid version of the baseline method called CRDR, which achieves a result of 0.837 in MRR on the CAsT-19 dataset. I wonder why the results of the hybrid search of CRDR are missing in Table 2.

**Reproducibility:**

3: Could reproduce the results with some difficulty. The settings of parameters are underspecified or subjectively determined; the training/evaluation data are not widely available.

**Reviewer Confidence:**

3: Pretty sure, but there's a chance I missed something. Although I have a good feel for this area in general, I did not carefully check the paper's details, e.g., the math, experimental design, or novelty.

---

### Official Review · Reviewer_Leu2 · 2023-08-04

**Soundness:** 3

**Excitement:**

2: Mediocre: This paper makes marginal contributions (vs non-contemporaneous work), so I would rather not see it in the conference.

**Missing References:**

Conversational query reformulation (CQR)
• [Can You Unpack That? Learning to Rewrite Questions-in-Context] (Elgohary et al., EMNLP-IJCNLP  2019)
• [Query and Answer Expansion from Conversation History.] (Yang et al., TREC 2019) Conversational Search
• [Open-Retrieval Conversational Question Answering] (Qu et al., SIGIR 2020)
• [Making Information Seeking Easier: An Improved Pipeline for Conversational Search] (Kumar &  Callan, EMNLP 2020)
• [Improving Conversational Passage Re-ranking with View Ensemble] (Ju et al., SIGIR 2023)


**Paper Topic And Main Contributions:**

The main contribution of this paper lies in enhancing the effectiveness of Conversational Search through the use of a novel framework based on Language Model Leveraging (LLM). The framework is designed to robustly represent user intents by generating multiple queries and hypothetical responses. Additionally, the paper introduces several approaches for prompting the Language Model Leveraging and aggregating the generated contents in experimental settings.

The proposed framework is evaluated using TREC CAsT benchmarks, showcasing the efficacy of the methods presented. Overall, the paper's major contribution lies in its empirical applications of employing LLM to enhance the searching capabilities in Conversational AI.

**Questions For The Authors:**

[Sec. 2] It is essential to clarify the distinction between "Generative IR" and "Text generation for IR" in the paper. I recommend referencing "Gen-IR at SIGIR'23" [https://arxiv.org/pdf/2306.02887.pdf] for more detailed insights on this particular research topic.

[Sec. 3.3.1] The paper mentions that the RAR method produces "M" hypothetical responses. However, it does not elaborate on how the response vectors (r_11, r12, ..r1M) are aggregated into a single new vector "s." Providing further explanation in this section would enhance the clarity of the method.

[Sec. 4.5] The comparison between RAR and RTR methods is limited since the RAR experiment seems to produce only one hypothetical response. To make the conclusion more robust, I suggest conducting experiments with the following settings: (i) N=1 vs. N=5(REW) and (ii) N=5, M=1(RTR) vs. N=5, M=5.

[Sec. 5] It is crucial to outline the experimental setup for T5QR, especially regarding its consideration of canonical responses. To ensure fairness, T5QR should be evaluated while taking into account responses in previous turns.

[Overall] Despite the declared limitation on computational costs and efficiency, it would be insightful to provide information about model parameter sizes or approximate numbers (e.g., times, FLOPs, model size, etc.). This information would help readers understand the trade-offs between different methods.

[Overall] To strengthen the paper's contributions, it would be valuable to clarify what the prompting-LLM approach can achieve compared to previous fine-tuned methods. Highlighting the advantages and disadvantages of this approach would offer a better understanding of its practical implications and potential in the field.

**Reasons To Accept:**

The paper explored how LLM can provide fine-grained content/knowledge (e.g., rewritten queries, hypothetical responses) and further validated the effectiveness through benchmark datasets. The author conducted a few experiments with various empirical settings for using prompts; in addition, they conducted a few simple qualitative analyses with human evaluations on LLM-generated content.

**Reasons To Reject:**

The paper's proposed methods lack fair and comprehensive comparisons. There is a notable absence of insights and takeaways concerning LLM-generated content, such as qualitative and empirical analyses or correlations between the generated content and improved effectiveness. Moreover, the paper fails to adequately justify the motivation behind using LLM to generate rewrites or responses. The absence of evidence demonstrating how the LLM-generated content outperforms previous rewriting approaches like T5QR is concerning.

Additionally, the approach of prompting-LM generation appears impractical for Conversational Search, and alternative strategies, such as reranking models, could be explored to improve retrieval effectiveness. For efficiency, supervised sparse retrieval methods could be investigated as well. Furthermore, the paper might benefit from considering how LLM can address more knowledge-intensive tasks like Conversational QA rather than focusing solely on Conversational Search.

In summary, the paper requires significant improvements in terms of comparative analysis, insightful findings regarding LLM-generated content, and justifications for its approach. Exploring alternative strategies for retrieval effectiveness and efficiency could enhance the paper's contributions to the field. Additionally, expanding the scope of application to knowledge-intensive tasks may lead to more impactful outcomes.

**Reproducibility:**

4: Could mostly reproduce the results, but there may be some variation because of sample variance or minor variations in their interpretation of the protocol or method.

**Reviewer Confidence:**

4: Quite sure. I tried to check the important points carefully. It's unlikely, though conceivable, that I missed something that should affect my ratings.

**Typos Grammar Style And Presentation Improvements:**

[Section 3.3] To enhance clarity and ease of understanding, I recommend rearranging the three aggregation methods as follows: "Max" (3.3.1) should come first, followed by "Mean" (3.3.3), and finally "Self-consistency" (3.3.2). This arrangement in an easy-to-hard order might improve the fluency of the section.

[Table 2; Sec. 4.2] It is crucial to ensure consistency in reporting. Many previous works mentioned in the related works section are not reflected in Table 2. To provide a comprehensive comparison, all relevant previous works should be included in Table 2, aligning with the scope and focus of the paper.

---

### Meta-Review · Area_Chair_QvnN · 2023-09-19

**Recommendation:** 4

**Metareview:**

Reviewers have mixed feelings about this paper.  The third review feels that the simple method is a novel contribution and the first review has a low excitement about this, both with high reviewer confidence.  Ultimately, LLM4CS is a new, if not unexpected, use of large language models and the authors argue that their empirical results are a significant contribution in the rebuttal.

---

### Decision · Program_Chairs · 2023-10-07

**Decision:**

Accept-Findings

**Comment:**

Reviewers have mixed feelings about this paper.  The third review feels that the simple method is a novel contribution and the first review has a low excitement about this, both with high reviewer confidence.  Ultimately, LLM4CS is a new, if not unexpected, use of large language models and the authors argue that their empirical results are a significant contribution in the rebuttal.